# The Feasibility of a Training Program for Peers with Severe Mental Illness to Provide One-to-One Services in Taiwan: A Pilot Study

**DOI:** 10.3390/ijerph19159124

**Published:** 2022-07-26

**Authors:** Kan-Yuan Cheng, Chia-Feng Yen

**Affiliations:** 1Department of Psychiatry, Taipei Veterans General Hospital, Yuli Branch, Yuli Township, Hualien 98142, Taiwan; 2Department of Public Health, Tzu-Chi University, Hualien 97071, Taiwan; mapleyeng@mail.tcu.edu.tw

**Keywords:** peer support, case management, independent living skill

## Abstract

Background: In Taiwan, services provided by patients’ peers in the mental health care system are still lacking. Therefore, this study aimed to develop a community setting model by a training program for severe mental illness (SMI) patients’ peers that also have SMI in Taiwan. Method: This training program comprised of 13-h lectures, 15-h practice classes, and an eight-week internship. In 2018 and 2019, the trainees provided one-to-one services to service users with SMI during the internship at a halfway house. The satisfaction and outcomes among all participants were measured in this training course. Results: The total mean satisfaction score in the training course for trainees (10 items, *n* = 13) and internship services for service users (12 items, *n* = 29) were 4.7 ± 0.4 and 4.6 ± 0.5, respectively. Among the trainees, 11 demonstrated improved Brief Psychiatric Rating Scale-18 (BPRS-18), Chinese Health Questionnaire-12 (CHQ-12), and Global Assessment of Functioning (GAF) scores after the whole training course. Among the 29 service users, their scores in the BPRS-18 and CHQ-12 decreased, whereas their scores in the GAF increased significantly under the internship service. Conclusion: In this pilot study, the trainees and service users who received internship services felt satisfied. The service providers and users with SMI both showed better clinical outcomes.

## 1. Introduction

A proportion of people with severe mental illness (SMI), including schizophrenia, bipolar disorder, severe major depressive disorder, delusional disorder, and organic mental illness, have deteriorated social or occupational functions, which could be related to positive or negative psychotic symptoms, impaired cognition, anxiety, or depression in the progress of mental illness [1,2]. Possible community barriers to reclaiming function among persons with severe mental illness include social discrimination and lacking support [3]. The neurocognition and internalized stigma are also associated with real-life functioning or quality of life among persons with SMI [4,5]. The concepts of recovery are emphasized in the modern mental health care system to overcome these disadvantages. The recovery-oriented services focus on personal preference and strength to promote functioning or clinical remission in daily life through empowerment [6].

A peer support service is one of the recovery-oriented services for persons with SMI [7]. In several developed countries, trained peers could deliver support or help to service users with mental illness in the specific case management services or vocational rehabilitation facilities [8,9]. Previous studies revealed that peer support services were effective in decreasing psychiatric symptoms, hospitalization, or emergency service use, while enhancing hope, empowerment, and social integration for persons with mental illness [10,11]. However, a meta-analysis reported equivocal results from services provided by peers compared with the usual care [12]. The diversity of service settings and absence of compatible control groups under blinded assessments in past studies of peer support services may explain such inconsistent results in term of analyzing group data [12,13]. On the other hand, the peer support worker could benefit from cultivating a hopeful attitude to recovery, knowledge, better social communication or work skills, and mutual support after completing the training program or providing peer support services based on the internship [14,15].

In Taiwan, there were 71 community rehabilitation centers, with a total capacity of 3406 persons (cover rate = 14.5/100,000), and 159 halfway houses, with a total of 6789 beds (cover rate = 28.8/100,000), as of December 2020. The number was still below the capacity of hospital treatment (22,690 beds, cover rate = 96.3/100,000) supported by National Health Insurance [16]. Because of the payment regulations of National Health Insurance, these interventions in the community settings for persons with mental illness are all directed by professionals. The medical model predominates service delivery in the country, and thus, difficulties in the realization of and self-determination in the principle of empowerment remain.

Peer support services could be a positive tipping point for community mental health care. In the previous studies for Chinese populations [14,15], the satisfying services are reported by service users with SMI who were provided with company in community activities or telephone counseling by a trained peer. The above studies’ results were based on a detailed description of the program’s process and a satisfactory questionnaire survey among service users. However, the clinical and functional outcomes among service users or trained peers are rarely discussed in these culture-relevant studies on Chinese populations.

A training course for peers has been organized to be integrated into case management services in halfway houses or supported housing programs in Taiwan. The purpose of this study was to examine the feasibility of a training program for peer support workers. Outcome variables were collected by surveying the satisfaction of trainees in the training course and service users receiving services who were paired up with designated trainees. We hypothesized that (1) trainees and service users would be satisfied with the whole training program, and (2) both sides would demonstrate better clinical and functional outcomes after completing a training program designed for the current study.

## 2. Materials and Methods

### 2.1. Study Participants

Between January 2018 and December 2019, a two-phase pilot study was conducted at a halfway house where 194 residents with SMI lived. This halfway house is provided by Taipei Veteran General Hospital, Yuli Branch, located in a rural area of eastern Taiwan [17]. We also recruited 23 persons with SMI who participated in a supported housing program in the area. Phase I involved organizing the training course for peers with SMI to be the assistants of case managers in the halfway house or supported housing program. Phase II of the study was held to invite other residents to receive internship services provided by the trained peers who cooperated with case managers under the supervision of professionals. We recruited all participants through posters on bulletin boards at the halfway house and oral invitations through the case manager in the supported housing program.

The inclusion criteria for both phases of the program were (i) having disability certifications with diagnoses of SMI in the social welfare system or mentally catastrophic illness in the health insurance system; (ii) living at a halfway house with a stable course of illness; (iii) being aged 20 years or older; and (iv) having a reading ability equivalent to 6 years of education. The exclusion criteria were (i) the comorbidity of severe physical illnesses which might result in hospitalization; (ii) acute exacerbation of psychosis; or (iii) being legally incompetent. For Phase I, there was one additional inclusion criterion: expressing willingness to be an assistant to case managers. Based on the regulations in the formal health care system in Taiwan, the diagnoses of severe mental illnesses, including schizophrenia, bipolar disorder, severe major depressive disorder, delusional disorder, and organic mental illness, were selected in the inclusion criteria [18]. Because of the low potential to be a peer support worker, the persons with diagnoses of the senile and childhood origin of mental illnesses were excluded.

We recruited prospective participants in Phase I and II through posters at the halfway house and in the community rehabilitation center two months before the initiation of data collection. The speech for invitation was held twice in the lobby of the halfway house, too. The research team applied for approval from the Institute Review Board of Antai Tian-Sheng Memorial Hospital (#19-036-A in 2019; #20-020-A in 2020) and received permission to collect data. All participants were asked to sign a written consent form and provided an opportunity to ask questions and decline participation.

### 2.2. Training Program for Trainees

There were two previous trainees and seven professionals across six medical disciplines discussing the framework and content of the training course for peer support workers in the focus group (Figure 1). Then, the course was held for trainees to be the intern assistances to case managers. Table 1 details the topics of the course utilizing principles of recovery and operations of peer support in case management services. The various intervention levels, based on the range of social connections in each topic, guided the teachers to emphasize the resource source in the peer support services (Table 1). The trainees participated in the course, which included 1 or 2 classes per week and lasted 16 weeks, including 13 h in lecture classes and 15 h in practice classes. At the end of the course, a final examination was held. Each trainee had to participate in all classes and pass the final exam in order to enter the next internship training.

Phase II of the study primarily comprised of internship practices. Participants’ changes in the outcome measures were assessed by comparing pre- and post-intervention score differences. All trainees were capable of providing a one-to-one service to the service users under supervision. The trainees had to practice in one 1-h session of practice every week at least for each service user. The caseload of trainees was 1:2 to 1:3 in the internship. The internship practice lasted for eight weeks, and the services included interviews to understand each other; storytelling to motivate the setting of goals in everyday life; organizing an independent living plan in the community under mutual agreement; and company, support, and tangible help in the practice of the plans. The teachers supervised trainees in the practice of the internship. In the first and last two sessions of the internship training, the teachers provided direct observations, evaluations of the service process, post-session discussions, and a final summary with feedback. The requirements of peer support services were summarized as a checklist in supervision. Because the potential candidates for peer workers are still unknown in Taiwan, we compared the characteristics and initially measured outcomes between the trainees and service users.

### 2.3. Measurements

#### 2.3.1. Social Support

We used the Social Support Scale (SSS) to measure participants’ perceived social support [18]. The instrument demonstrated good reliability for the current study. Internal consistency was also confirmed by a Cronbach’s α value of 0.86. The Kaiser–Meyer–Olkin (KMO) value of 0.82 and the Bartlett test of sphericity (BT) of 815.37 (*p* < 0.001) in factor analysis revealed the construct validity of the scale [19]. This scale contains three dimensions: social support from relatives or family (SSS-R), friends or peers (SSS-F), and staff or professionals (SSS-S). Higher scores indicate greater social support.

#### 2.3.2. Mental Health and Psychiatric Symptoms

The Chinese Health Questionnaire-12 (CHQ-12) was employed to assess the mental health of service users and trainees. The value of the area under the Receiver Operating Characteristic curve was 0.85, and the cutoff value was 3/4 [20]. The sensitivity of the questionnaire was 78%, and the specificity was 77% [19]. The lower the score, the better the subject’s mental health is. We also used the Brief Psychiatric Rating Scale-18 (BPRS-18) to measure the psychiatric symptoms of the service users and trainees. The split-half reliability of BPRS was 0.96–0.98 in the Taiwanese population [21]. The test and retest reliability (interclass correlation coefficient values = 0.73–0.91) were adequate [21]. A low CHQ-12 score suggests less severe psychiatric symptoms.

#### 2.3.3. Functioning

The Global Assessment of Functioning (GAF) was administered to assess participants’ overall functions influenced by their mental illness conditions. Jones et al. reported a reliability coefficient of 0.72 [22]. The Chinese version of the Social Functioning Scale (C-SFS) was also employed to measure subjective social function by Song, who adapted the Social Functioning Scale considering cultural factors in Taiwan [23]. The internal consistencies are acceptable to good, with a Cronbach’s α value of 0.86 for the scale and a Cronbach’s α value ranging from 0.48 to 0.88 for the subscales [23]. The higher the scores, the better the respondents’ self-reported functions are.

### 2.4. Process of Assessment

The self-report questionnaires, SSS, CHQ-12, and C-SFS, were administered by an occupational therapist (MPK) assistant. The subjective BPRS-18 and GAF scales were surveyed by a board psychiatrist (KYC) who had experience in international multiple-center clinical trials. Both the assistant and investigator were members of the intervention group. There was no control group or blind procedure in this study considering its exploratory nature.

### 2.5. Statistical Methods

We used the independent *t*-test to examine the indicators, which were continuous variables between groups, and the chi square test for categorical variables in the comparisons between the groups of trainees and service users. If the above continuous variable did not meet a normal distribution, the Mann–Whitney U test was selected. Differences between pre- and post-intervention were analyzed by a paired *t*-test; the Wilcoxon sign’s rank test was used for the comparisons with small sample sizes. The Wilcoxon sign’s rank test was also used to assess continuous variables in which the assumption of normal distribution was violated. International Business Machines Corporation, New You, NY, USA, Statistical Product and Service Solutions (SPSS) Statistics 16.0 was employed for all statistical analysis.

## 3. Results

Originally, there were 13 trained workers participating in the training program initially. Two of them withdrew after passing the final exam but declined to enter the internship. A total of 31 service users participated in the internship services, and two of them discontinued their participation in Phase II of the study because they were unwilling to continue. At the end of the study, no subjects struggled with acute psychotic symptoms, suicidal thoughts, or homicidal risk. All participants persistently lived in the community during the study period.

### 3.1. Characteristics and Initially Measured Outcomes of Study Participants

Table 2 shows the characteristics of the trainees and service users. The trainees and service users were middle-aged, and there were slightly more female members. The trainees had significantly more academic years than the service users (15.7 ± 2.9 vs. 12.5 ± 4.0, df = 40.3, t = 2.94, *p* < 0.01). The majority of the trainees and service users were unmarried, and a larger proportion of them lived in the halfway house than in the supported housing program. Regarding participants’ psychiatric history, the trainees and service users both had major diagnoses of schizophrenia. The onset age, duration of mental illness and hospitalization, and proportion of past suicide or homicide behaviors all were not significantly different in the comparisons. The trainees had significantly higher scores for SSS-R (57.8 ± 7.6 vs. 49.9 ± 14.6, df = 39.7, t = 2.34, *p* = 0.03), GAF (67.9 ± 8.3 vs. 60.8 ± 7.4, df = 42, t = 2.81, *p* < 0.01), and C-SFS (77.7 ± 8.5 vs. 65.9 ± 15.3, df = 42, t = 2.60, *p* = 0.01) than the service users at the pre-intervention stage but not other variables.

### 3.2. The Satisfaction of the Training Course for Trainees

As shown in Table 3, the total mean satisfaction score of the training course for trainees was 4.7 ± 0.4. The mean scores of the Overall, Self-growth, and Experiences ranged from 4.6 to 4.9, 4.7 to 4.7, and 4.3 to 4.8, respectively.

### 3.3. The Satisfaction of Internship Service for Service Users

Table 4 shows the satisfaction scores towards internship services, which were rated by the service users who completed the whole training course. The total mean score was 4.6 ± 0.5. Scores of Interaction, Experience, and Reputation ranged from 4.2 to 4.9, 4.0 to 4.7, and 4.6 to 4.8, respectively.

### 3.4. The Changes in Measured Outcomes in Trainees

The trainees reported an increased score on the Social Support Scale, though the change was not statistically significant (43.7 ± 8.2 vs. 47.5 ± 8.0, df = 10, z = 1.62, *p* = 0.11). There were insignificant changes in the total and other two dimension scores of the Social Support Scale among the trainees (Table 5). However, the scores for the CHQ-12 (7.1 ± 5.1 vs. 1.3 ± 2.1, df = 10, z = −2.21, *p* = 0.03) and BPRS-18 (32.7 ± 4.7 vs. 26.3 ± 1.3, df = 10, z = −2.81, *p* = 0.01) among the trainees both decreased significantly after the whole training course. The trainees also had remarkable increased scores for GAF, but few changes to the C-SFS score at the end of the training course (Table 5).

### 3.5. The Changes in Measured Outcomes in Service Users

Although not significant, the service users reported increased scores for the total SSS and all three dimensions (see Table 6). The diminished score for CHQ-12 was revealed in the service users after one-to-one internship practice (5.3 ± 4.4 vs. 1.6 ± 2.0, df = 28, z = −3.24, *p* < 0.01). The service users showed significantly decreased scores for BPRS-18 (36.3 ± 6.1 vs. 30.1 ± 3.9, df = 28, t = −6.41, *p* < 0.01) and increased GAF scores (60.4 ± 6.6 vs. 67.9 ± 8.3, df = 28, t = 6.09, *p* < 0.01). There were insignificant changes to the C-SFS scores among the service users under the internship services.

## 4. Discussion

### 4.1. Main Findings in This Pilot Study

In this pilot study, the good level of satisfaction in the training course for trainees and the internship service for service users could demonstrate the satisfied feasibility of training program (Table 3 and Table 4). The low rate of withdrawals (*n* = 4, 9%) and no major adverse events during the study period confirm the safety of the training course. Both the trainees and service users with SMI diagnosed with schizophrenia (*n* = 41, 93.2%) showed greater positive changes in their mental health (CHQ-12), psychiatric symptoms (BPRS-18), functioning, and their daily lives (GAF) compared to their counterparts. However, the social support (SSS) and social functioning (C-SFS) among all participants did not improve significantly in the survey.

### 4.2. The Satisfied Acceptability of the Whole Training Course

In this pilot study, the 13 trainees felt satisfied with the total of 22 classes in the training course (mean ± sd of the satisfaction score is 4.7 ± 0.4). These findings replicate a previous study in Hong Kong, in which 18 peer support workers reported satisfactory scores ranging from 3.7 to 4.3 after 24-week paid internship training, which required them to provide telephone services between August 2011 and December 2012 [14]. In a vocational peer support training program in Hong Kong, their six trainees also reported mean satisfactory scores ranging from 4.0 to 4.8 after completing the curriculum, which consisted of 15-session coursework, eight-session storytelling workshops, and a 50-h practicum from February 2014 to January 2015 [24].

The 29 service users also reported satisfying feelings in the one-to-one services (Table 4). These findings are comparable with an earlier study conducted in China, in which face-to-face services in community activities were provided to 54 consumers with SMI by 13 trained peers [15]. A high proportion of satisfied feelings (*n* = 39, 72.2%) was reported among the 54 consumers [15].

However, the eight-week internship training in our study was shorter than the 24-week practicum under supervision in the telephone peer service program used in the Hong Kong study [14]. By contrast, the Phase I training program in our study consisted of 22 classes, similar to the 23 sessions in the other Hong Kong study for supported employment peer services [24]. Unlike past studies, our participants were not monetarily compensated, because most of them work a part-time job under supported employment or received work training in workshops.

In the context and contents of the training program for peers, the introduction of peer support and concept of recovery, issue of stigma or discrimination, recovery storytelling, ethics and role of the peer support worker, communication and other work skills, and strength or self-determination under empowerment are common in the Phase I study (Table 1) and the previous earlier study in Hong Kong [24]. The contents seem to be the effective elements in the training program, suggesting continuous use for peers with SMI in the Chinese population for future research.

### 4.3. The Candidates for Trainees and Outcomes among Peers under the Whole Training Course

In this pilot study, the recruited trainees had relatively higher educational levels, better social support from family or relatives, and better global functions and self-reported social functions than the volunteers of the service users. However, two of the 13 trained peers were unwilling to provide one-to-one service during the internship. The rationale reported was that they felt pressure in the one-to-one interactions, and the internship practice did not include compensation. Therefore, further study could consider inviting persons with SMI who have graduated from university (mean ± sd of academic year = 15.7 ± 2.9 years), moderate to high family support (mean ± sd of SSS-R score = 57.8 ± 7.6), global functions with mild influence by mental illness (mean ± sd of GAF score = 67.9 ± 8.3), and moderate to high independence in social functions (mean ± sd of C-SFS score = 77.7 ± 8.5) to be candidates for peer support worker training.

After the completion of training, the trainees in this pilot study had better mental health, better control of psychiatric symptoms and global function, and were less impacted by mental illness (Table 5). In an earlier study of peer support service in China, 11 of 13 peer service workers with SMI were willing to provide services after the project [15]. Based on a semi-structured questionnaire survey, a great number of the 13 peer workers reported improvements in working skills (76.9%), social communication skills (61.5%), and mood (53.8%) after providing assistance to 54 consumers participating in community activities [15]. The above findings by Fan et al. could partially support our results of improved mental health and clinical outcomes among trainees who completed the training course.

### 4.4. The Outcomes among Service Users by Peer-Delivered Intervention in the Internship

Findings of the current study are consistent with past systematic reviews, in which peer support workers’ better psychiatric symptom control and decreased admission with longer tenure in the community for persons with SMI were commonly reported after similar programs [11,25]. In earlier meta-analysis studies, Fuhr et al. analyzed data from 14 studies, all from high-income countries The study authors reported that peer-delivered intervention improved the clinical symptoms among persons with severe mental illness (SMD = 0.14, 95%CI = 0.57 to 0.29, *p* = 0.51, I^2^ = 0%, *n* = 84), equivalent to mental services provided by professional services [12]. In a recent Australian study, Hancock et al. used qualitative data from 82 questionnaires from 58 individual in-depth interviews. They found that most of their participants with SMI were discharged from a hospital; returned to their lives and daily routines more easily; were more capable of obtaining new knowledge, strategies, and skills; and developed a sense of hope through peer-delivered services [26]. Similarly, better mental health was observed by our participants who received the service provided by trainees.

However, no significant changes in social support and social function were found in our study. This finding is incompatible with previous meta-analysis study results, in which 23 studies with 3329 participants in total indicated the opposite [27]. The article by White et al. found possible benefits of social support but no impact on clinical symptoms after peer-delivered services among persons with SMI [27]. An emphasis on self-management of mental illness in the trainees’ internship course might explain our study’s positive clinical outcomes. The issues of managing mental illness in everyday activities occupied the largest proportion of the total of 236 sessions of one-to-one visits (*n* = 105, 44.5%). With close supervision by professionals, the trainees participating in the training course might improve their capacities to support the service users to learn more appropriate skills or better strategies for coping with psychiatric symptoms.

Counterintuitively, the service users did not report significantly better social support and functions in this pilot study. The lower frequency of company in activities or tangible help in the service sessions (*n* = 28, 11.9%) might be related to the lower impact on social support or social functions among the service users. Moreover, participants might need more time to cultivate new skills, form habits, or modify behaviors, and therefore, eight sessions for each service user might insufficient. Additionally, the improvement in symptomatology and clinical severity of mental illness have significantly predicted the enhancement of real-world function among the 72 persons with schizophrenia who participated in computer-assisted cognitive remediation and social skills training in an earlier study [2]. Therefore, more service sessions with longer durations and facilitating company or help may be considered for designing a future internship course for trainees with SMI.

### 4.5. Study Limitations

Lacking a satisfaction questionnaire for trainees in the internship resulted in unknown trainees’ satisfaction under professional supervision. Furthermore, raters were not blind to the experimental conditions, and therefore, possible biases might have contaminated their ratings. Finally, the lack of a control group and small sample size in this study led to limitations in the interpretation of outcome changes.

## 5. Conclusions

The high level of agreement in satisfaction among trainees and service users demonstrates the utility of this training course, including the 13-h lectures, 15-h practice classes, and an eight-week internship for trainees with severe mental illness to be assistants to case managers. The improved mental health and clinical outcomes in trainees and service users with severe mental illness in this study suggest benefits of peer-delivered interventions as an addition to case management services in community mental health care in Taiwan. Because there were no significant impacts on social support and social functions in this pilot study, a longer intervention period and the promotion of a company or tangible help in peer support services for persons with severe mental illness could be considered in Taiwan. Monetary compensation for trainees and assistants to case managers after completing training could also be arranged under most developing peer support services for persons with severe mental illness in Taiwan. Finally, one of the most important steps for future research will be utilizing larger-scale randomized controlled studies.

## Figures and Tables

**Figure 1 ijerph-19-09124-f001:**
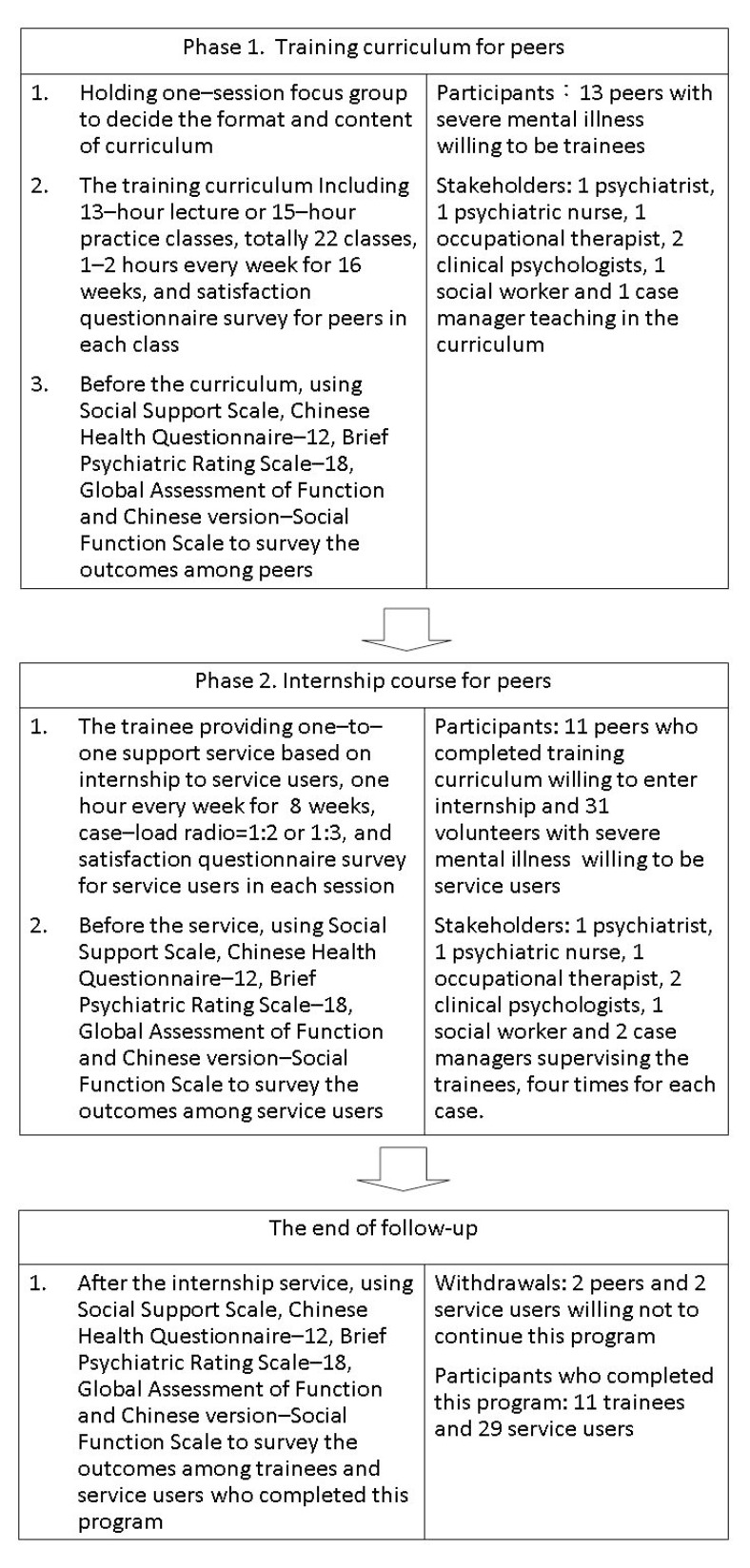
The process of this pilot study.

**Table 1 ijerph-19-09124-t001:** The training course for trainees with severe mental illness.

Topics	Intervention Level		
	Community	Service System	Individual
We need peer support	Introducing the People with Disabilities Rights Protection Act in Taiwan *	The needs for independent living in the community *	What is peer support? *
	Dealing with discrimination encountered in communities *		
Well-being first	Using resources in communities to cultivate independent living skills, better health, and skills to find a job *	Well-being and mental health promotion *	The experiences of self-management in mental health care *
		Treatment and rehabilitation in mental health services *	
My recovery journey	Practice resource connection in your life circle **	Concept of recovery and empowerment at both individual- and service-team-levels *	Operation of self-determination *
			My life story **
Recovery together	The practice of planning for independent living in the community **	Roles of trained professionals with different specialties and peer support workers in mental health services *	Planning for independent living in the community **
			The encouragement at critical time points for promoting motivation *
			Life storytelling ***
Work together		Communication with the other members of the service team *	Interview: Empathy and non-judgmental attitude ***
		Cooperation with others in independent living planning **	Interview: active listening skills **
			Interview: decision making under mutual agreement ***

* 1-h lecture, ** 1.5-h practice, *** 2-h practice.

**Table 2 ijerph-19-09124-t002:** The characteristics, psychiatric history, and measured outcomes among trainees and service users.

	Trainees (*n* = 13)		Service Users (*n* = 31)		Comparisons		
	*n*	(%)	*n*	(%)	Statistics	df	*p* Value
Demography							
Age (mean ± sd)	51.1	±11.5	48.9	±8.8	t = 0.69	42	0.49
Sex							
Male	6	(46.2)	13	(41.9)	χ^2^ = 0.07	1	0.80
Female	7	(53.8)	18	(58.1)			
Education years (mean ± sd)	15.7	±2.9	12.5	±4.0	t = 2.94	30.4	<0.01 *
Marriage							
Unmarried	13	(100)	27	(87.1)	χ^2^ = 1.86	2	0.40
Married	0	(0)	2	(6.5)			
Divorced	0	(0)	2	(6.5)			
Housing							
Halfway house	10	(76.9)	27	(87.1)	χ^2^ = 0.71	1	0.40
Supported housing services	3	(23.1)	4	(12.9)			
Comorbidity of chronic physical illness	7	(53.8)	16	(51.6)	χ^2^ = 0.02	1	0.89
Psychiatric history							
Psychiatric diagnoses							
Schizophrenia	12	(92.3)	29	(93.5)	χ^2^ = 0.82	2	0.66
Bipolar disorder	1	(7.7)	1	(3.2)			
Organic mental illness	0	(0)	1	(3.2)			
Onset age (mean ± sd) ^a^	24.2	±9.6	25.0	±9.8	z = −0.35	42	0.73
Duration of hospitalization (mean ± sd) ^a^	10.4	±10.3	15.8	±9.7	z = −0.93	42	0.06
Duration of mental illness (mean ± sd) ^a^	26.9	±11.5	23.9	±9.9	z = −1.89	42	0.35
History of suicide or homicide	7	(53.8)	12	(38.7)	χ^2^ = 0.86	1	0.36
Measured outcomes							
Social Support Scale—Total (mean ± sd)	162.9	±13.9	151.5	±37.8	t = 1.46	41.6	0.15
Family and relatives (mean ± sd)	57.8	±7.6	49.9	±14.6	t = 2.34	39.7	0.03
Peers and friends (mean ± sd)	45.4	±8.5	47.0	±15.2	t = −0.37	42	0.72
Professionals and staff (mean ± sd)	59.7	±7.9	54.5	±12.6	t = 1.37	42	0.18
Chinese Health Questionnaire-12 (mean ± sd) ^a^	7.0	±5.1	5.7	±4.6	z = 0.99	42	0.32
Brief Psychiatric Rating Scale-18 (mean ± sd)	33.7	±4.9	36.3	±5.9	t = −1.39	42	0.17
Global Assessment of Functioning (mean ± sd)	67.9	±8.3	60.8	±7.4	t = 2.81	42	<0.01 *
Chinese version—Social Function Scale (mean ± sd)	77.7	±8.5	65.9	±15.3	t = 2.60	42	0.01 *

^a^ Mann–Whitney U Test, * *p* < 0.05.

**Table 3 ijerph-19-09124-t003:** The satisfaction of the training course for trainees (*n* = 13, 22 classes).

Items	Satisfaction Score (1–5)
Overall	
I feel that the time arrangement of the class is appropriate	4.8 ± 0.3
I can understand the whole contents of the class	4.6 ± 0.6
I feel that the teacher was friendly to me in teaching	4.9 ± 0.3
I have no interest in this class (opposite counting)	4.3 ± 0.6
Self-growth	
I have already learned the new skills	4.4 ± 0.5
I have already gotten useful knowledge	4.7 ± 0.4
I believe I can help others more after this class	4.7 ± 0.5
Experiences	
I can feel that everyone made an effort to learn in the class	4.8 ± 0.4
I feel easy and free in the class	4.5 ± 0.3
The experiences that the others shared in the class can help me	4.8 ± 0.4
Total	4.7 ± 0.4

**Table 4 ijerph-19-09124-t004:** The satisfaction of services based on internship for service users (*n* = 29, 236 sessions).

Items	Satisfaction Score (1–5)
Interaction	
I feel that the peers are friendly to me when interacting	4.9 ± 0.3
I feel that the peer can understand my thinking	4.7 ± 0.4
I feel free and without pressure when interacting with the peers	4.7 ± 0.5
I feel that the peers can understand my feelings	4.7 ± 0.4
I feel pressure or uneasy when interacting with the peers (opposite counting)	4.2 ± 0.6
Experience	
I feel that the experience that the peer shared is helpful to me	4.7 ± 0.6
I can speak my ideas out loud when interacting with the peers	4.7 ± 0.5
I can do things differently after interaction with the peers	4.6 ± 0.7
I feel that the services provided by the peers are not useful to me (opposite counting)	4.0 ± 0.8
Reputation	
When I have difficulties, I will talk to peers in the future	4.7 ± 0.3
I will continue my participation in this service	4.8 ± 0.5
I will invite others to participate in this service	4.6 ± 0.5
Total	4.6 ± 0.5

**Table 5 ijerph-19-09124-t005:** The measured outcomes among trainees before and after the whole training course (*n* = 11).

	Before		After		df	Z ^a^	*p* Value
	Mean	±sd	Mean	±sd			
Social Support Scale—Total	160.6	±14.0	164.6	±27.9	10	1.07	0.29
Family and relatives	58.0	±8.3	58.4	±14.0	10	0.36	0.72
Peers and friends	43.7	±8.2	47.5	±8.0	10	1.62	0.11
Professionals and staff	58.9	±8.3	58.7	±9.9	10	−0.71	0.48
Chinese Health Questionnaire-12	7.1	±5.1	1.3	±2.1	10	−2.21	0.03 *
Brief Psychiatric Rating Scale-18	32.7	±4.7	26.3	±1.3	10	−2.81	0.01 *
Global Assessment of Functioning	69.0	±8.6	76.5	±6.3	10	2.30	0.02 *
Chinese version—Social Function Scale	77.4	±9.1	78.3	±8.7	10	0.77	0.44

^a^ Wilcoxon sign’s rank test, * *p* < 0.05.

**Table 6 ijerph-19-09124-t006:** The measured outcomes among service users before and after the internship services (*n* = 29).

	Before		After		df	t	*p* Value
	Mean	±sd	Mean	±sd			
Social Support Scale—Total	153.5	±37.4	158.0	±33.6	28	0.82	0.42
Family and relatives	50.4	±14.8	52.6	±14.2	28	094	0.35
Peers and friends	47.6	±15.3	50.5	±13.1	28	1.30	0.20
Professionals and staff	55.5	±12.0	54.9	±13.4	28	0.32	0.75
Chinese Health Questionnaire-12 ^a^	5.3	±4.4	1.6	±2.0	28	z = 3.24	<0.01 *
Brief Psychiatric Rating Scale-18	36.3	±6.1	30.1	±3.9	28	−6.41	<0.01 *
Global Assessment of Functioning	60.4	±6.6	67.9	±8.3	28	6.09	<0.01 *
Chinese version—Social Function Scale	66.1	±16.0	68.4	±17.7	28	0.84	0.41

^a^ Wilcoxon sign’s rank test, * *p* < 0.05.

## Data Availability

Not applicable.

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
