# Peer review of "The Feasibility of a Training Program for Peers with Severe Mental Illness to Provide One-to-One Services in Taiwan: A Pilot Study"

_ijerph, 2022, doi:10.3390/ijerph19159124_

Round 1
Reviewer 1 Report
The present study explores the effects of a peer-led training program for people with severe mental illness conducted in Taiwan. Peers providing the training as well as participants receiving the intervention improved both in the severity of symptoms and overall quality of life.
The study is timely and interesting, particularly considering the conflicting results available in literature regarding the efficacy of peer-led interventions in different mental disorders. The manuscript is, overall, well-structured and the analyses appear to be appropriate.
The paper shows some relevant methodological limitations, but they are reported and discussed, albeit briefly, in the appropriate section.
However, some important issues have to be addressed to improve the clarity of the manuscript
General issues:
-Some typos and grammatical errors are present, and some sentences are not very clear and difficult to read. It is advisable to submit the manuscript to professional proofreading or, at least, to have it checked by a native English speaker.
Abstract:
-More details regarding the included sample should be provided in the Abstract. For instance, at least the number of peers providing the interventions and of participants and the recruitment area should be reported also in this section.
Introduction:
-A definition of severe mental illness should be provided to enhance clarity for the reader, taking into account the multidisciplinary nature of the Journal. In particular, different definitions of this condition are reported in literature: usually severe mental illnesses include schizophrenia spectrum disorders, bipolar disorders and severe major depressive disorders, but some Authors extend this category to other debilitating mental disorders as well.
-The aims and the main hypotheses of the present study should be better defined and reported. It would be advisable to add a subparagraph dedicated to the aims of the study at the end of the Introduction section.
Methods:
-Relating to first comment regarding the Introduction section, better description of the inclusion criteria related to severe mental illness is required for international readers to better understand the included sample and the reference population.
-More details should be provided regarding statistical analyses: in particular, it should be specified how the normality of data distribution was checked and which non-parametric tests were used with non-normally distributed variables; the latter can be derived from the tables in the results section, but they should also be reported in the statistical analyses subsection.
Discussion:
-Recent evidence suggest that evidence-based psychosocial interventions are more effective when combined with other evidence-based treatments or are integrated into structured rehabilitation programs (see Nibbio et al., Evidence-Based Integrated Intervention in Patients with Schizophrenia: A Pilot Study of Feasibility and Effectiveness in a Real-World Rehabilitation Setting. Int J Environ Res Public Health. 2020;17(10):3352. doi:10.3390/ijerph17103352 and Vita et al., Effectiveness, Core Elements, and Moderators of Response of Cognitive Remediation for Schizophrenia: A Systematic Review and Meta-analysis of Randomized Clinical Trials. JAMA Psychiatry. 2021;78(8):848-858. doi:10.1001/jamapsychiatry.2021.0620): more discussion should be provided regarding the need to combine peer-led interventions with other evidence-based interventions, particularly in more clinically compromised participants.
-The main limitations of the present study are reported in the appropriate section, but more discussion is currently required. For instance, the lack of a control group represents an important issue, but it could be resolved in future research based on the results of the present study.
-The size of the sample should also be discussed: while some significant effects were observed on key study outcomes, the small sample size could be regarded as a prominent limitation of the work. Was a power analysis conducted prior to study enrollment in order to determine the required sample size? If so, this detail should be reported in the manuscript.
Author Response
Revision Note
Manuscript ID: ijerph-1719572
Type of manuscript: Article
Title: The feasibility of a training program for peers with severe mental illness to provide one-to-one services in Taiwan: a pilot study
Dear Reviewer,
We would like to thank you for the valuable suggestions. We have revised the manuscript based on your reminding and related to the full-text content in marked up using the “Track Changes” function of Word. Our responses and the pagination for the revisions in the revised manuscript are listed as follows with a point-by-point response and appendix to the reviewers’ comments.
Sincerely
Kan-Yuan Cheng
- General issues: Some typos and grammatical errors are present, and some sentences are not very clear and difficult to read. It is advisable to submit the manuscript to professional proofreading or, at least, to have it checked by a native English speaker.
(1) We have invited a native English speaker to correct the errors in grammar and edit the unclear sentences.
- Abstract: More details regarding the included sample should be provided in the Abstract. For instance, at least the number of peers providing the interventions and of participants and the recruitment area should be reported also in this section.
(2) The recruitment facility has been added in this section (page 1, line 17). Because the numbers of peers and service users had mentioned in part of results, we did not repeat it.
- Introduction: A definition of severe mental illness should be provided to enhance clarity for the reader, taking into account the multidisciplinary nature of the Journal. In particular, different definitions of this condition are reported in literature: usually severe mental illnesses include schizophrenia spectrum disorders, bipolar disorders and severe major depressive disorders, but some Authors extend this category to other debilitating mental disorders as well.
(3) In this study, the diagnoses of severe mental illness based on the definition in formal health care system in Taiwan. We have reported it in the introduction section and paragraph of inclusion criteria (page 1, line 32-33 and page 4, line 105-110).
- The aims and the main hypotheses of the present study should be better defined and reported. It would be advisable to add a subparagraph dedicated to the aims of the study at the end of the Introduction section.
(4) In the end of introduction section, the aims and main hypotheses have been reported (page 2, line 77-80).
- Methods: Relating to first comment regarding the Introduction section, better description of the inclusion criteria related to severe mental illness is required for international readers to better understand the included sample and the reference population.
(5) The definition of severe mental illness in this study has been mentioned in detail under the regulation of formal health care system in Taiwan (page 4, line 105-110).
- More details should be provided regarding statistical analyses: in particular, it should be specified how the normality of data distribution was checked and which non-parametric tests were used with non-normally distributed variables; the latter can be derived from the tables in the results section, but they should also be reported in the statistical analyses subsection.
(6) The statistical processes have been described in detail. The authors also decided to use Wilcoxon sign’s rank test to exam the changes of outcomes among the trainees because of small number (n=11) (page 6, line 192 to 199, page 9, Table 5).
- Discussion: Recent evidence suggest that evidence-based psychosocial interventions are more effective when combined with other evidence-based treatments or are integrated into structured rehabilitation programs (see Nibbio et al., Evidence-Based Integrated Intervention in Patients with Schizophrenia: A Pilot Study of Feasibility and Effectiveness in a Real-World Rehabilitation Setting. Int J Environ Res Public Health. 2020;17(10):3352. doi:10.3390/ijerph17103352and Vita et al., Effectiveness, Core Elements, and Moderators of Response of Cognitive Remediation for Schizophrenia: A Systematic Review and Meta-analysis of Randomized Clinical Trials. JAMA Psychiatry. 2021;78(8):848-858. doi:10.1001/jamapsychiatry.2021.0620): more discussion should be provided regarding the need to combine peer-led interventions with other evidence-based interventions, particularly in more clinically compromised participants.
(7) We have added the article, reported by Nibbio et al. in the reference list and used the main finding of this reference to explain about the potentially functional improvement among service users in our study (page 12, line 357 to 360 and reference list).
- The main limitations of the present study are reported in the appropriate section, but more discussion is currently required. For instance, the lack of a control group represents an important issue, but it could be resolved in future research based on the results of the present study.
(8) The further study with the better designs has been suggested in the final part of conclusion (page 13, line 382 to 383).
- The size of the sample should also be discussed: while some significant effects were observed on key study outcomes, the small sample size could be regarded as a prominent limitation of the work. Was a power analysis conducted prior to study enrollment in order to determine the required sample size? If so, this detail should be reported in the manuscript.
(9) We have changed the statistical method in analyzing the variables with small sample size (page 9, Table 5). The further study with the larger scale has been suggested in the final part of conclusion (page 13, line 382 to 383).

Reviewer 2 Report
The topic of the study is for real interest for services offered for patients with SMI.
The introduction can be improved by adding research about other intervention programs, related to the topic.
The methodology and design are appropriate, but the number of participants are small.
The results are in accordance with the used instruments.
I suggest another presentation for the satisfaction questionnaires (Table 3 and Table 4).
I suggest, also, an improvement for the discussion of the main results, in line with the introduction.
Author Response
Revision Note
Manuscript ID: ijerph-1719572
Type of manuscript: Article
Title: The feasibility of a training program for peers with severe mental illness to provide one-to-one services in Taiwan: a pilot study
Dear Reviewer,
We would like to thank you for the valuable suggestions. We have revised the manuscript based on your reminding and related to the full-text content in marked up using the “Track Changes” function of Word. Our responses and the pagination for the revisions in the revised manuscript are listed as follows with a point-by-point response and appendix to the reviewers’ comments.
Sincerely
Kan-Yuan Cheng
- The introduction can be improved by adding research about other intervention programs, related to the topic.
(1) The statement of benefits among peer support worker under training or providing services has been added in the introduction. The feasibility studies of training program for peers with severe mental illness have been also reported (page 2, line 51 to 54 and 65 to 71).
- The methodology and design are appropriate, but the number of participants are small.
(2) We have used the Wilcoxon sign’s rank test to exam the changes in the variables with small sample size (page 9, Table 5). The further study with the better design has also been suggested in the final part of conclusion (page 13, line 382 to 383).
- I suggest another presentation for the satisfaction questionnaires (Table 3 and Table 4).
(3) Because the sentences of questions in Table 3 and Table 4 are not short, it would be not easy to be read in the figure presentation. We have used the colored columns in the all Tables and Figure to be clearly read (Figure 1, Table 1 to 6).
- I suggest, also, an improvement for the discussion of the main results, in line with the introduction.
(4) In line with improved introduction, the whole paragraph of 4.2 has been rewritten (page 10, line 278 to page 11, line 304). And, supplement of evident in discussion about functional improvement under intervention in persons with schizophrenia (page 12, line 357 to 360).

Reviewer 3 Report
Thank you for the opportunity to review this interesting and important study. Overall you have provided a clear description of the study and the results.
There are some English language grammatical problems that I expect will be addressed in production
The introduction should end with the aim of the study clearly stated. There seems to be 3 - mental health, clinical and functional outcomes for trainees and service users if the intervention was expected to change these (secondary outcomes?) was there a hypothesis that functioning or mental state would improve because of participating in the program?
- satisfaction of trainees with the course (phase 1) and
-satisfaction of the service users with the trainees (phase 2)
2.1 study participants - how is severe mental illness defined and how were study participants recruited? the inclusion and exclusion criteria is in the next section but this does not explain the process of recruitment how did interested people get involved after they saw the posters?
2.2 discussing the framework and content of the training course - should the word be designing not discussing?
low rate of withdrawals from training and peer support is a very good outcome.
In some countries and services, peer workers hold paid positions. This potential could be raised in the discussion.
Author Response
Revision Note
Manuscript ID: ijerph-1719572
Type of manuscript: Article
Title: The feasibility of a training program for peers with severe mental illness to provide one-to-one services in Taiwan: a pilot study
Dear Reviewer,
We would like to thank you for the valuable suggestions. We have revised the manuscript based on your reminding and related to the full-text content in marked up using the “Track Changes” function of Word. Our responses and the pagination for the revisions in the revised manuscript are listed as follows with a point-by-point response and appendix to the reviewers’ comments.
Sincerely
Kan-Yuan Cheng
- There are some English language grammatical problems that I expect will be addressed in production
(1) We have invited a native English speaker to correct the errors in grammar and edit the unclear sentences.
- The introduction should end with the aim of the study clearly stated. There seems to be 3 - mental health, clinical and functional outcomes for trainees and service users if the intervention was expected to change these (secondary outcomes?) was there a hypothesis that functioning or mental state would improve because of participating in the program? satisfaction of trainees with the course (phase 1) and satisfaction of the service users with the trainees (phase 2)
(2) In the end of introduction section, the aims and main hypotheses have been reported (page 2, line 77-80).
- 2.1 study participants - how is severe mental illness defined and how were study participants recruited? the inclusion and exclusion criteria is in the next section but this does not explain the process of recruitment how did interested people get involved after they saw the posters?
(3) The diagnoses of severe mental illness accorded to the definition in formal health care system in Taiwan. We had proposed it in the introduction section and paragraph of inclusion criteria (page 1, line 32-33 and page 4, line 105-110). The process of recruitment has been described in the paragraph of inclusion criteria (page 4, line 108-111).
- 2.2 discussing the framework and content of the training course - should the word be designing not discussing?
(4) We have rewritten the 4.2 paragraph of discussion about training course by comparing the results of satisfaction survey between this pilot program and previous studies for Chinese population (page 10, line 278 to page 11, line 304). Also, the context and content of training course has been discussed (page 11, line 298-304).
- In some countries and services, peer workers hold paid positions. This potential could be raised in the discussion.
(5) We have proposed the reason of payment for trainees in internship in the future (page 11, line 296-297), and suggested that these trained peers could be the paid assistant to case manager in Taiwan (page 12, line 380-382).

Round 2
Reviewer 1 Report
The Authors have answerd all queries in a satisfactory manner.
The manuscript has been consistently improved.